# General practitioners' perspectives on statutory skin cancer screening–A questionnaire-based cross-sectional survey in Germany

**Lydia Reinhardt**[1,2]*, **Cristin Strasser**[1,2], **Theresa Steeb**[3,4], **Anne Petzold**[3,4], **Markus V. Heppt**[3,4], **Anja Wessely**[3,4], **Carola Berking**[3,4‡], **Friedegund Meier**[1,2‡]

1 Department of Dermatology, University Hospital Carl Gustav Carus, TU Dresden, Dresden, Germany,
2 Skin Cancer Center at the National Center for Tumor Diseases (NCT/UCC), Dresden, Germany,
3 Department of Dermatology, Uniklinikum Erlangen, Friedrich-Alexander-Universität Erlangen-Nürnberg, Erlangen, Germany, 4 Comprehensive Cancer Center Erlangen-European Metropolitan Area of Nürnberg (CCC ER-EMN), Erlangen, Germany

☯ These authors contributed equally to this work.
‡ CB and FM also contributed equally to this work.
* Lydia.Reinhardt@ukdd.de

**Data Availability Statement:** All relevant data are within the manuscript and its Supporting Information files (dataset, questionnaire and categories from the qualitative analysis).

## Abstract

### Background

In Germany, skin cancer screening (SCS) is available free of charge every two years to all those with statutory health insurance over the age of 35. General Practitioners (GP) can carry out the screening if they have completed an 8-hour training course. GPs play a crucial role in the implementation of SCS and act as gatekeepers between initial patient contact and referral to dermatologists.

### Objective

To record how comprehensively GPs carry out SCS in terms of patient information and body examination, as well as to explore GPs opinions on the feasibility of SCS.

### Methods

A cross-sectional survey was conducted. A questionnaire was sent to GPs with permission to perform SCS in two regions of Germany (Bavaria and Saxony) between August and September 2021. Data were analyzed using descriptive analysis. Subgroup analysis was performed according to regions (federal state, location of physician´s office), professional experience (experience in years, number of monthly screenings, age) and gender. Open questions were evaluated using qualitative content analysis.

### Results

In the survey, 204 GPs responded. Genitalia (40.7%, 83/203), anal fold (62.3%, 127/204) and oral mucosa (66.7%, 136/204) were the least examined body regions during screening.

**Funding:** The Skin Cancer Center at the University Hospital Dresden received a grant for material resources from the Foundation Stiftung Hochschulmedizin Dresden, project ID 50176 (https://stiftung-hochschulmedizin.de/). The funders had no role in study design, data collection and analysis, decision to publish, or preparation of the manuscript.

**Competing interests:** The authors have declared that no competing interests exist.

Information on risks (false-positive findings: 18.6%, 38/203; false-negative findings: 13.2%, 27/203; overdiagnosis: 7.8%, 16/203) and benefits (48.0%, 98/202) were not always provided. GPs who performed screenings more frequently were more likely to provide information about the benefits of SCS (p<0.001; >10 vs. <5 screenings per month). Opinions were provided on uncertainties, knowledge requirements, structural and organizational requirements of SCS, SCS training and evaluation. The organization and remuneration of the SCS programme was seen as a barrier to implementation. GPs expressed uncertainties especially in unclear findings and in dermatoscopy.

## Conclusion

Uncertainties in the implementation of the SCS should be addressed by offering refresher courses. Good networking between GPs and dermatologists is essential to improve SCS quality.

## Introduction

The incidence of skin cancer has been increasing worldwide since the 1970s. The cases of non-melanoma skin cancer (NMSC) and melanoma together make skin cancer the most frequently diagnosed type of cancer in Germany [1]. Rising life expectancy and increased exposure to ultraviolet radiation through leisure activities are discussed as possible reasons for the rising incidence of skin cancer [2, 3].

The earlier skin cancer is detected, the greater the chances of cure, especially in cases of melanoma that are highly likely to metastasize [4]. In Germany, a national skin cancer screening programme (SCS) was introduced in 2008 [5]. During screening, the entire skin is examined for conspicuous pigment spots and skin lesions. Everyone with health insurance over the age of 35 is entitled to a fully reimbursed screening every two years. Dermatologists and other physicians such as general practitioners (GP) are allowed to carry out the examination, provided they have completed a required 8-hour training course. This training includes information on benefits and harms of early detection measures, clinical pictures and etiology, frequency and risk factors of skin cancer. The training is designed to ensure that physicians can perform standardized visual whole-body skin examination, including scalp and intertriginous areas [6]. The content is standardized according to the guideline for early cancer detection [7]. The program is offered to both dermatologists and other disciplines [8]. Since 2020, the use of a dermatoscope is also reimbursed as part of the SCS [9]. Screening as a preventive service is aimed at people who have not yet developed skin cancer. If a GP suspects a suspicious skin lesion, the patient is referred to a dermatologist for a second screening examination [10]. Nevertheless, the benefit is discussed controversially. A long-term reduction in melanoma mortality has not yet been demonstrated [11–14]. There is also limited direct evidence regarding the negative effects of screening [15]. However, SCS could lead to an overdiagnosis of melanoma [16]. Instead, there is intensive discussion about risk-stratified screening, i.e., whether people with an increased risk of skin cancer should be screened preferentially [17].

GPs play a crucial role in the implementation of SCS as they act as gatekeepers between initial patient contact and referral to dermatologists [18, 19]. In 2018, over two-thirds of GPs in Saxony (72,37%, 1.383/1.911) and Bavaria (68,89%, 7.072/10.265) and the majority of dermatologists in Saxony (93,85%, 183/195) and Bavaria (85,99%, 528/614) had permission to

perform SCS [20]. According to billing data in 2018, 218 377 SCS were billed by GPs in Saxony and 687 715 in Bavaria. In comparison, 239 598 SCS were billed by dermatologists in Saxony and 481 831 in Bavaria. Thus, screenings were performed with approximately equal frequency by GPs and dermatologists in Saxony, but more frequently by GPs in Bavaria. However, these data may not reflect the actual number of screenings, as other examinations were also billed under this billing code [20, 21]. Waiting times for an appointment with a dermatologist are often longer, especially in rural areas [19].

Previous studies revealed that the quality of skin examination and the preventive information given during SCS varies between dermatologists and other specialists, including GPs [22, 23]. One study on patient-reported quality of SCS showed that dermatologists were significantly more likely than other specialists to examine all body areas according to the screening programme's specifications [22]. Furthermore, screening participants reported receiving advice and written information on skin cancer prevention from dermatologists more frequently [22]. However, none of the studies investigated aspects from the perspective of GPs performing SCS. In international studies, GPs reported time constraints, competing patient comorbidities, and lack of training as the most frequent barriers to performing skin cancer examinations [24].

So far, little is known about how GPs themselves evaluate the national SCS programme. The aim of this survey was to assess how GPs conduct the screening and provide patient education in this context. We investigated whether there were differences between regions (federal state, location of physician´s office), professional experience (professional experience in years, number of monthly screenings, age) and gender. In addition, we explored the personal opinion of GPs about the SCS programme in general, as well as uncertainties and barriers to the implementation.

## Materials and methods

### Study design and participants

Within the project SaBaScreen (The quality of national skin cancer screening from the perspective of GPs: A questionnaire-based comparison between Saxony and Bavaria) we conducted a cross-sectional survey (see S1 Appendix: STROBE checklist). The study was performed by the Skin Tumor Center at the University Hospital Dresden (Saxony) and the Department of Dermatology at the University Hospital Erlangen (Bavaria). A vote of the ethics committee of the Technical University of Dresden dated 12.05.2021 is available (BO-EK-164032021).

The study region comprised the two large cities of Dresden (Saxony) and Nuremberg (Bavaria) and their neighbouring counties. A large city is defined as a municipality with at least 500,000 inhabitants [25]. Contact details of GPs with permission to perform SCS were obtained from the Associations of Statutory Health Insurance Physicians in Saxony and Bavaria (December 2020). All physicians practicing general medicine at that time who were authorized to perform SCS and were active in the study region were included and contacted individually by mail. The mailing included a covering letter, a study information outline, a paper-based questionnaire and a prepaid envelope for returning the questionnaire. GPs provided written informed consent to participate. A predefined ID was used to ensure that each GP contacted could only participate in the survey once. GPs did not receive a remuneration for their participation. From August 2021 to October 2021, a total of 1153 GPs were contacted, 543 in Bavaria (7 counties) and 610 in Saxony (5 counties). Fig 1 shows the recruitment process. The proportion of female GPs was 48.7% in the surveyed population, with Saxony having a much higher proportion of female GP at 58.5% compared to Bavaria at 37.6%.

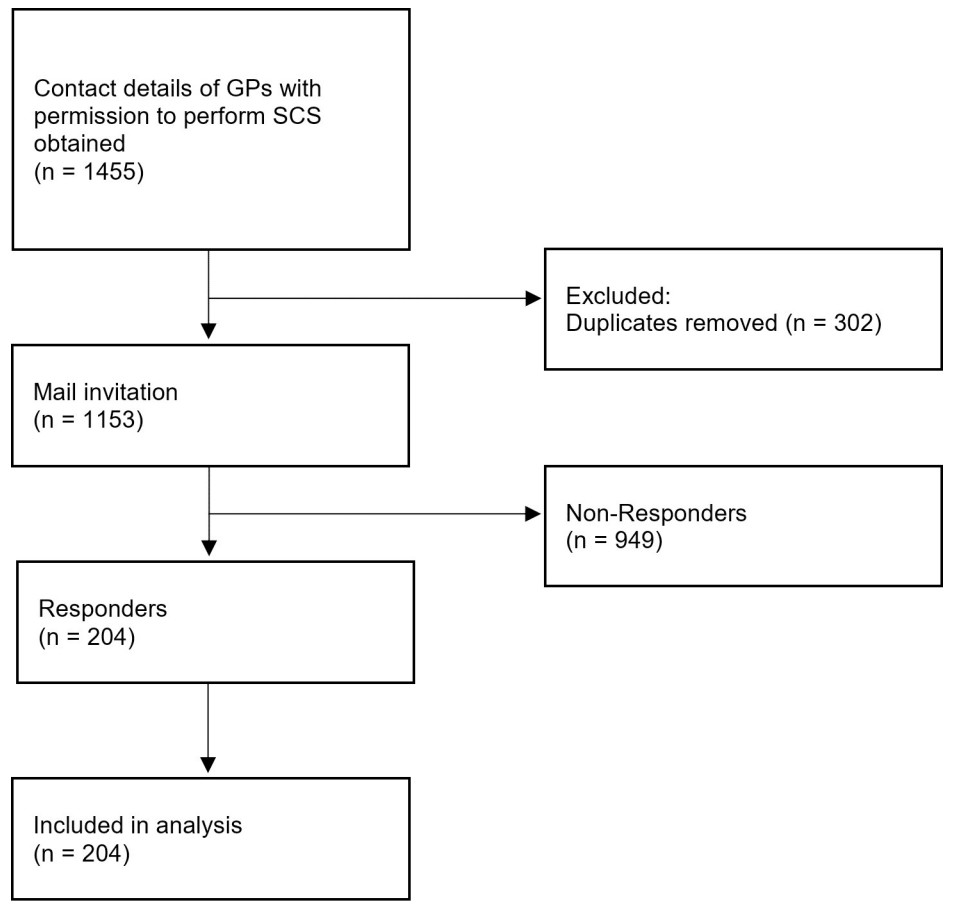

Abbreviation: SCS = skin cancer screening

**Fig 1. Recruitment flowchart.**

## Measures

We assessed GP´s performance on SCS and their personal attitude towards SCS using a questionnaire developed by the Skin Tumor Center at the University Hospital Dresden and the Department of Dermatology at the University Hospital Erlangen (S2 Appendix). As there was no validated instrument that met our study objective, we developed a questionnaire. Professionals with expertise in general medicine and dermato-oncology reviewed the questionnaire. Based on their suggestions, the questionnaire was revised to the final form.

The questionnaire consisted of 37 questions on different aspects of SCS, including four open questions. The first part inquired GPs for general information about SCS for patients, such as risks and benefits. We asked participants to rate statements like "I inform my patients as part of skin cancer screening about causes of skin cancer" and "I take a detailed medical history on skin cancer risk and ask about personal medical history". In accordance with the screening programme's specifications for the standardized visual whole-body skin examination, the second part asked in detail which parts of the body were examined during the screening, including scalp and intertriginous areas. These questions from the first two parts were to be answered on a five-point Likert scale as to whether individual aspects of SCS were *always*, *often*, *occasionally*, *rarely*, or *never* performed in the opinion of the GP. Following questions

on the need for regular training in SCS have been published elsewhere [26].The questionnaire also contained demographic data on age, gender, location of the physician´s office, and professional experience. In addition, open questions were asked on preferred time intervals of refresher courses, uncertainties when performing SCS and need of further knowledge. Participants could also provide further comments on the implementation of SCS. The answers to all open questions were analyzed using categories based on a qualitative content analysis. The survey dataset can be found in S3 Appendix.

## Data analysis

**Quantitative analysis.** Following the Consensus-Based Checklist for Reporting of Survey Studies (CROSS), data were analyzed by means of descriptive analysis and subgroup analysis. Responses were analyzed using software R, version 4.0.2 (https://www.r-project.org/). An estimated sample size of n = 86 per comparison group (federal state) was calculated by a two-sided two-sample t-test power calculation with an alpha error of α = 5%, a power of 1-β = 90%, a delta of Δ = 0.5 and a standard deviation of σ = 1. Subgroup analyses examined differences in federal state, location of physician´s office, professional experience, number of monthly screenings, gender, and age. Due to multiple testing, we adjusted alpha following the Bonferroni correction to a value of $\alpha_{adjusted} = \frac{\alpha}{6} = \frac{0.05}{6} = 0.0083$ and focused on those who *always* performed certain actions to obtain more robust results. We used $X^2$-tests for these analyses. Ordinally scaled data are given as percentages and absolute values. The rank numbers of the categories are:

$$R \left( X_i = \left\{ \begin{array}{c} never \\ rarely \\ sometimes \\ often \\ always \end{array} \right\} \right) = \left\{ \begin{array}{c} 1 \\ 2 \\ 3 \\ 4 \\ 5 \end{array} \right.$$

With $R$ being the rank, $X_i$ the $i_{th}$ observed value. The mean rank $\bar{R}$ of a subgroup is given as the arithmetic mean of ranks.

**Open question analysis.** The purpose of the open questions was to explore additional views and opinions that may not have been captured in the questionnaire. Following Mayring [27], the approach for the qualitative analysis was based on a combination of deductive and inductive methods. It was conducted by three researchers (CS, LR, and TS) with different professional backgrounds (health care management, sociology, and public health). Two of the main categories were predetermined by questions related to uncertainties and knowledge needs about the SCS training programme. Respondents also had the opportunity to add further comments. The categories were generated from all given answers. In a first step, the text material was transferred into Microsoft Excel. It was then reviewed and sorted by one of the researchers (CS). Subsequently, the identified text passages were generalized and transferred into a category system. The categories developed were reviewed by the research team (LR and TS) and revised if necessary.

## Results

### Quantitative analysis

**Study population.** Of 1153 questionnaires, 204 were returned (response rate 17.7%), of which 60.3% were GPs from Saxony (vs. 37.9% from Bavaria). The median age of participants was 55 years (mean = 53.71, range 34–77). The majority of respondents were female (59.3%, 121/204), and had more than 15 years of professional experience (59.8%, 122/204). 27.9%

**Table 1. Characteristics of study participants (n = 204).**

| | | Total (n = 204) |
|---|---|---|
| **Gender** | female | 121 (59.3%) |
| | male | 81 (39.7%) |
| | diverse | 2 (1.0%) |
| **Age** | Mean ± standard deviation | 53.71 ± 9.51 |
| | Median (range) | 55 (34–77) |
| **Location of physician´s office** | Large city | 57 (27.9%) |
| | City | 71 (34.8%) |
| | Rural area | 75 (36.8%) |
| | missing | 1 (0.5%) |
| **Duration of professional experience** | <5 years | 27 (13.2%) |
| | 5–15 years | 54 (26.5%) |
| | >15 years | 122 (59.8%) |
| | missing | 1 (0.5%) |
| **Amount of performed SCS/month** | <5 | 47 (23.0%) |
| | 5–10 | 80 (39.2%) |
| | >10 | 76 (37.3%) |
| | missing | 1 (0.5%) |

**Abbreviations:** SCS = skin cancer screening

work in an office in a large city (57/204), 34.8% (71/204) in a city, and 36.8% (75/204) in a rural area. According to self-report, 23% (47/204, missing = 1) perform less than 5 screenings per month. 39.2% (80/204, missing = 1) of respondents perform 5 to 10 screenings per month, and 37.3% (76/204, missing = 1) perform more than 10 (Table 1).

**Patient education on SCS and medical history.** About half of the respondents *always* gave information about prevention of skin cancer (51.5%, 105/204), benefits of SCS (48%, 98/204, missing = 2), and causes of skin cancer (44.1%, 90/204) (Fig 2). Information about potential risks was given less frequently. A proportion of respondents stated to *never* give information in the context of SCS about potential false-positive findings (15.2%, 31/204, missing = 1), potential false-negative findings (8.3%, 17/204, missing = 1), and a potential overdiagnosis (23%, 47/204, missing = 1). There were only 10 GPs always providing comprehensive information in the context of SCS, thus indicating that 95.1% (194/204) did not.Half of the participants reported *always* inquiring the patient's medical history (51.5%, 105/204, missing = 2). The majority *always* (41.2%, 84/204, missing = 3) or *often* (20.6%, 42/204, missing = 3) also obtained the family history.

**Examination of the individual body regions.** The GPs surveyed reported *always* examining the patient's upper body, arms, and legs (100% each, 204/204) (Fig 3). Predominantly, the head and scalp (91.2%, 186/204), ears (94%, 193/204), armpits/groin (96.1%, 196/204), hands (99%, 202/204), interdigital spaces (90.2%, 184/204, missing = 1), and feet (98%, 200/204) were *always* examined, according to the GPs.

Toe interspaces were *always* examined by 88.2% (180/204). The majority of respondents reported *always* (66.7%, 136/204) examining the mouth and oral mucosa. The mouth was *rarely* part of the screening in 7.4% (15/204) of the GP respondents and *never* in 1.5% (3/204).

The buttocks were *always* examined by 78.4% (160/204) and *often* by 12.3% (25/204). The anal fold, on the other hand, was *always* examined by only 62.3% (127/204). Overall, 6.4% (13/

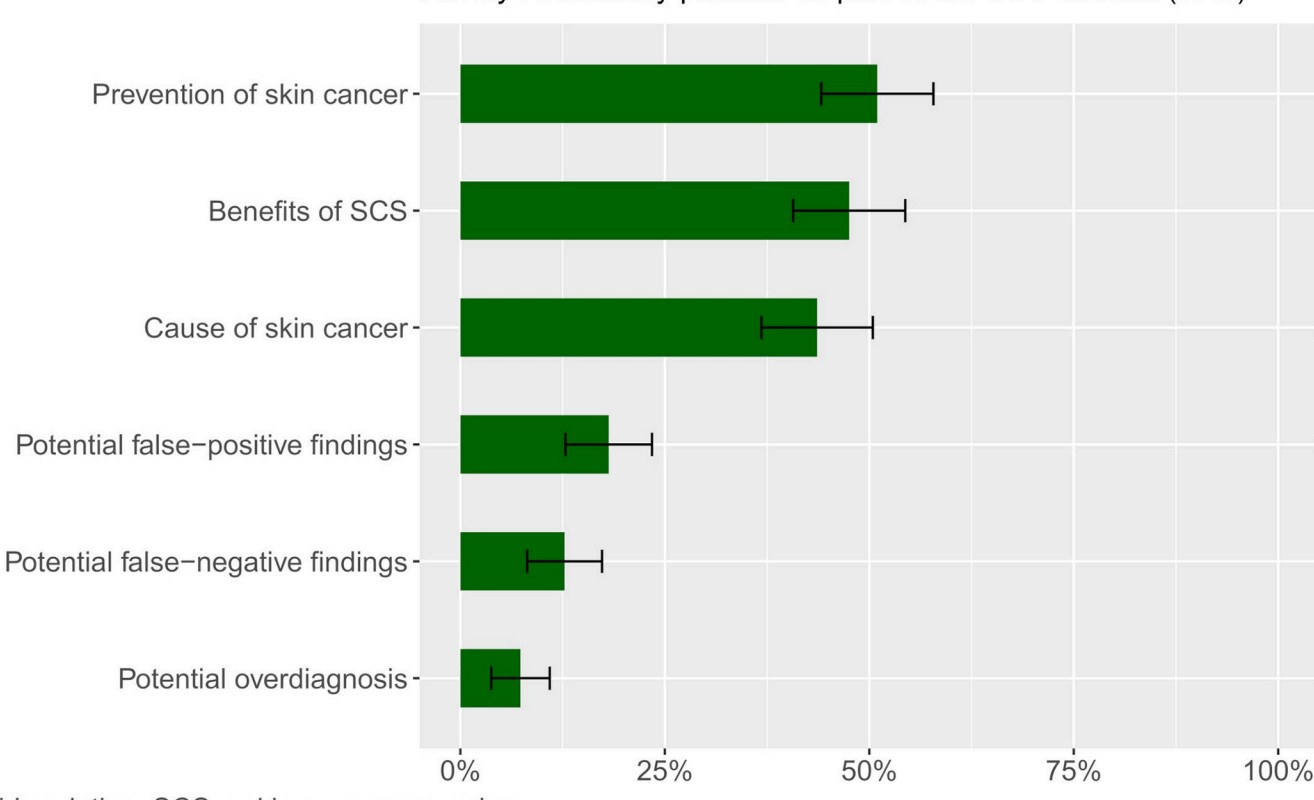

I *always* inform my patients as part of the SCS about... (in %)

Abbreviation: SCS = skin cancer screening

**Fig 2. Patient information that was *always* provided during SCS by general practitioners, in % (n = 204).**

204) *rarely* examined this body region and 3.9% (8/204) *never* examined it. Genitalia were the body region that most respondents reported screening *rarely* (14.2%, 29/204, missing = 1) or *never* (3.4%, 7/204, missing = 1). Less than half reported doing so *always* (40.7%, 83/204, missing = 1) or *often* (23.5%, 48/204, missing = 1). In summary, only 31.9% (65/204) *always* performed SCS in its entirety, whereas 68.1% did not.

**Subgroup analysis.** Few significant differences are apparent in the subgroup analysis. For significance, following the Bonferroni correction, the p-value of the $X^2$-test must fall below an alpha of 0.0083. GPs conducting >10 screenings per month *always* informed patients about the benefits of SCS more frequently (68.4%) compared to those conducting 5–10 screenings (40.0%) and those conducting <5 screenings per month (29.8%) (p<0.001). Additionally, the analyses revealed that GPs aged >53 years significantly more often *always* examined patients' genitals within the scope of SCS (50.9% vs 38.9%, p = 0.0026). No further significant differences could be detected.

## Open question analysis

A total of 94 GPs answered at least one open question. 63 of these were from Saxony. Six main categories with additional topics emerged from the analysis: (I) uncertainty when performing SCS, (II) knowledge required for SCS, (III) implementation of SCS, (IV) training on SCS, (V) structural and organizational requirements of SCS, and (VI) evaluation of SCS. Due to the large number of subcategories, here we focus only on those that relate to the content described in the

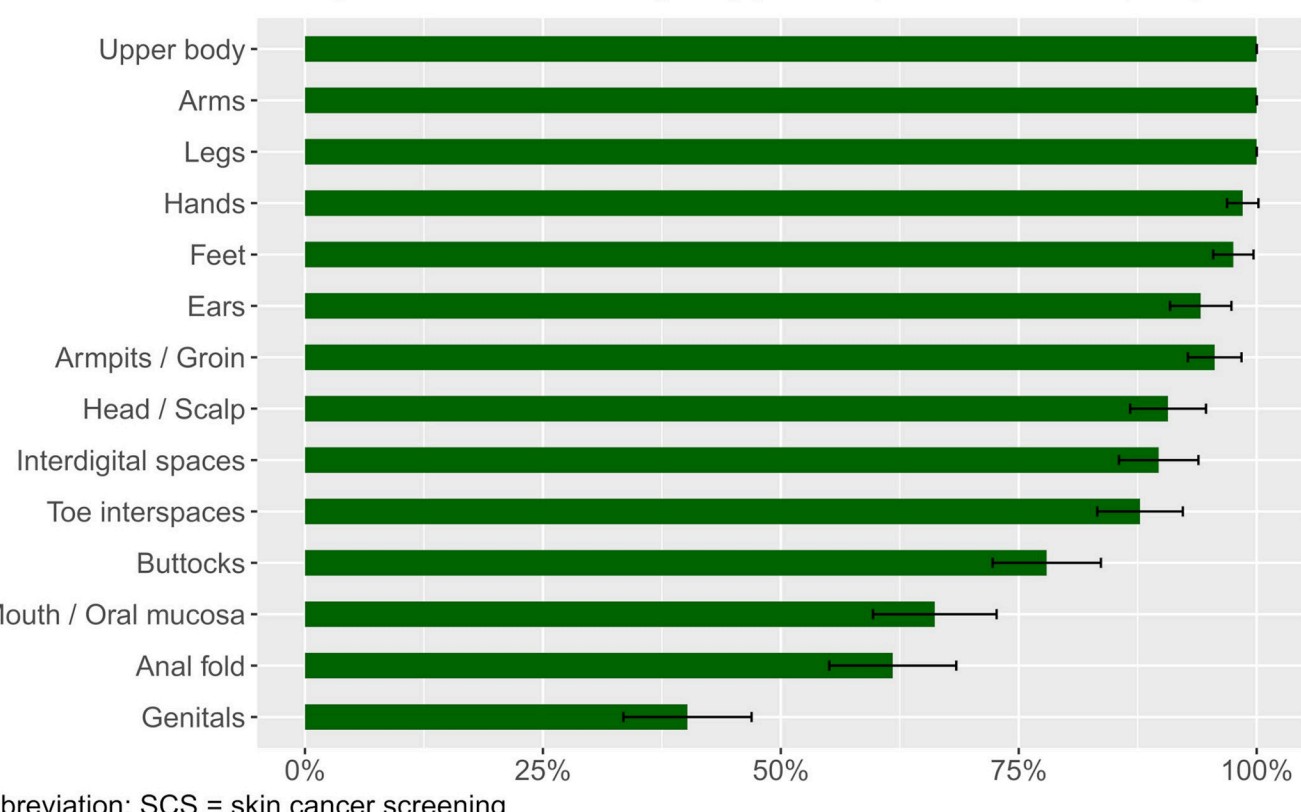

**Fig 3. Body sites that were *always* examined during SCS by general practitioners, in % (n = 204, missing = 2).**

quantitative part of the analysis. Categories are in summary form and supported by example citations. To ensure participant anonymity, any quotes with identifying information were modified. The categories referring to the training programme (II and IV) are not considered in detail here. The full set of categories and their frequencies can be found in S4 Appendix.

**Uncertainty when performing SCS.** Uncertainties in SCS were mostly attributed to a difficult differential diagnosis and a lack of practical experience or routine.

> „*It is very difficult for me to distinguish between dysplastic nevus and suspected melanoma.—I also sometimes have difficulties with the terminology of relevant skin lesions.—Perhaps that is why I refer patients to specialists more often than necessary.*"

Another challenge was the use of dermatoscopy. As of 2020, the provision of a dermatoscope is mandatory, but previously, using a dermatoscope was not part of the training. One GP stated that he attended a dermatoscopy course on his own for this reason. In contrast, two GPs found the use of the dermatoscope helpful in case of unclear findings.

Continuing education content was described as not relevant to practice. Visual teaching material and a focus on "unusual findings and unusual body sites" were desired. Further uncertainties referred to the communication between GP and dermatologist. Either there was no possibility to get a second opinion from a specialist (no dermatology office in the region) or there was no feedback after the patients´ referral to the dermatologist. GPs encountered

uncertainties with the referral of patients to the dermatology office. They criticized that referrals may be more frequent than necessary.

**Implementation of SCS.**   Only one of the GPs described asking patients to self-examine themselves as part of his routine at SCS. Four GPs considered the support of artificial intelligence or digital tools to be useful. Photo apps, online consultations and supervision of images via e-mail were mentioned.

*"A possibility to have skin pictures supervised via e-mail would provide much more certainty. In our region, there are no appointments with dermatologists available!"*

However, it was emphasized that Artificial Intelligence should not replace medical consultation.

**Structural and organizational requirements of SCS.**   There were statements that indicated insufficient remuneration for SCS compared to the effort.

*„SCS with sometimes more than 50 moles takes about 30 minutes. It is therefore completely inadequately remunerated."*

In contrast, only one GP stated that the payment seems appropriate.

*„Remuneration is ok—screening only takes between 5 and 10 minutes."*

Two GPs stated that they perform SCS or a skin examination as part of the general checkup examination. Under the "Checkup 35" scheme, people with statutory health insurance in Germany are entitled to a general checkup every three years from age 35. It was repeatedly noted that SCS as part of "Checkup 35" is unfavorably organized, since SCS can be billed every 2 years, but the "Checkup 35" only every 3 years.

Some of the GPs saw the responsibility for the genital examination with gynecologists or urologists.

*„The demand to also inspect women's genitalia during skin cancer screening is illusory in general medical office."*

Lack of timely appointments with dermatologists and lack of cooperation with dermatologists were also cited as problems.

**Evaluation of SCS.**   Opinions on the potential benefits of screening varied widely. From one GP´s perspective, abnormal findings are also readily detected outside of the screening. Reference was made to a US study that failed to demonstrate an effect of SCS on mortality. One GP considered an evaluation of SCS to be necessary. If the benefit is proven, regular refresher courses should be offered. Another GP was concerned that the nationwide screening of people under 25 years of age, as currently promoted by health insurance companies, could lead to an increase in false-positive findings.

## Discussion

SCS is considered an important tool for early detection of skin cancer. In Germany, all persons over 35 years with statutory health insurance, who have not previously developed skin cancer, can take advantage of the SCS every two years. Dermatologists can perform the SCS as well as other specialists who have completed an 8-hour training course. GPs play an important role as they are often the first physician contact [18, 19]. However, various studies imply that there

might be relevant differences in the quality of SCS performed by non-dermatologists. Najmi et al published a systematic literature review of documented barriers to diagnostic skin cancer examinations reported by primary care providers, patients, and health systems [24]. Lack of time, competing patient comorbidities, and lack of training in performing skin cancer screenings were the most frequently reported barriers for primary care providers. A German study described the patient-reported quality of SCS showing differences in the quality of skin examinations and preventive information between dermatologists and other physicians. [22]

The views of GPs in Germany who are authorized to perform SCS following a training course have not been explored in detail yet. Therefore, the aim of our study was to evaluate GPs' perspectives on SCS in Germany, in terms of implementation, patient education, and other issues relevant to implementation to identify barriers and potential approaches to overcome those barriers. For this purpose, GPs in Saxony and Bavaria provided information in a questionnaire on how they perform the screening and what they inform their patients about. They also had the opportunity to address other aspects of the SCS in open questions. Education about prevention and causes of skin cancer are essential to counteract the development of skin cancer [28]. The majority of respondents reported providing this information (Fig 2). Since there is no clear evidence on the benefit of SCS in terms of mortality reduction [11–14], it is important to educate patients about possible false-positive and false-negative findings or a potential overdiagnosis. However, our results showed that GPs interviewed only provide this information occasionally or not at all (95.1%). One reason for this lack of information could be limited time or other routine examinations. It is also possible that this aspect is hardly addressed in the training courses. In the open questions, patient education was mentioned only once. However, education about skin cancer and guidance on skin self-inspection can raise awareness on skin cancer. Well-informed patients may seek timely medical attention when appropriate.

More professional experience was not associated with better patient education in our survey. On the other hand, GPs who performed screenings more frequently provided information about benefits of SCS more often (p<0.001; >10 vs. <5 screenings per month). Possibly, GPs value SCS as a more important preventive measure the more frequently they perform screenings. Furthermore, GPs aged >53 years were significantly more likely to always examine patients' genital area during SCS (50.9% vs 38.9%, p = 0.0026). However, due to the small sample, these statistically significant results of our subgroup analysis should be interpreted with caution. Further research with a larger sample would be desirable here.

Furthermore, we asked GPs how regularly they examine certain body regions during screening. Our results showed that the GPs always examine easily visible body regions such as the arms, face and trunk. Body regions that are more difficult to see, such as spaces between the toes, oral mucosa and anal fold, are not always examined, according to the GP´s self-report (Fig 3). Gender and age seem to have an impact on how the screening is performed [29, 30]. The examination of the genital region in particular was considered unrealistic in a GP examination. These results are consistent with previous studies on SCS from the perspective of patients [22]. Skin lesions in these areas are rather rarely and more difficult to detect, and the examination of the genital area and the anal fold can be shameful [31]. Our results revealed that 68.1% of GPs did not always examine all body areas during SCS. Awareness for skin cancer and good cooperation, such as prompt referral to a specialist, are therefore essential. Patients should be informed during the consultation that melanomas can also grow on the mucosa; and in women, for example, abnormalities should be discussed with the physician during a regular gynecological checkup [31]. Qualitative analysis also showed that the time required for screening was perceived very differently. Therefore, varying screening quality must be expected. A study of patient-reported quality of SCS also showed that the duration of

screening varies widely. Dermatologists screened all body areas more often and provided information about prevention more frequently than other specialists [22].

Other important outcomes of the open-ended questions, in which GPs could address further relevant topics, were uncertainties with differential diagnoses, lack of practical experience or routine and the use of the dermatoscope [32]. Participants wanted these topics to be addressed in the mandatory 8-hour training programme, especially the use of the dermatoscope. They also wanted refresher courses to be offered and more opportunities for continuing training. Even though the use of the dermatoscope is already part of the training course, there seems to be a need for further training in its practical application. Personal attitudes toward continuing education and the desire for more advanced offerings were also examined in our survey and published elsewhere [26].

Participants reported several barriers to the implementation of the SCS, mainly organizational and structural requirements, such as a lack of cooperation between dermatologists, the lower payment in the context of the "Checkup 35" programme and long waiting times for an appointment with the dermatologist. Some GPs also expressed skepticism about the benefits of the screening programme, especially when predominantly people without an increased risk of skin cancer are screened. This ties in with discussions about risk-adapted screening that prioritizes those who are at increased risk of developing skin cancer, such as people with bright skin and multiple naevi [33]. These structural barriers and the lack of evidence may be important reasons for GPs not to perform SCS.

Especially in rural areas where it is difficult to get a dermatologist appointment, GPs play an important role in the implementation of the SCS and in detection of skin cancer at an early stage [18, 19]. While previous studies have shown that patients prefer to see a dermatologist for screening [19, 22, 34], waiting times for an appointment with a GP are often shorter than with dermatologists [19, 22]. In addition, screening by GPs may reach a different target group than screening by dermatology specialists only [35]. Therefore, the role of GPs at the SCS should be strengthened by offering regular training courses. An evaluation of the existing training courses is therefore needed to optimize the quality of the screening in the long term [13, 26]. A well-developed network between the doctors is desirable, which enables queries about abnormal findings and the secure exchange of referrals and medical reports.

## Strength and limitations

In this study, we examined GPs' perspectives on SCS in Germany. By evaluating open questions, we were able to gather additional information and ascertain which topics on SCS were highlighted positively or negatively by GPs. These results are valuable indications of how the screening programme and the training courses can be improved in the long term.

The response rate was relatively low, which might indicate a low relevance of the topic from the GP's point of view, since SCS may not have the highest priority among other examinations. Presumably, GPs critical of SCS did not participate. Overall, more GPs from Saxony than from Bavaria responded (60.3% vs. 39.9%). This is also reflected in the results of the qualitative content analysis, in which more GPs from Saxony answered at least one of the open questions. The reason might be the coordination of the survey from the Skin Tumor Center at Dresden so that GPs from Saxony felt more addressed. In the subgroup analysis, the target sample size of n = 86 per group was not always achieved, so it is possible that significant differences were missed. Especially the comparison of GPs performing a different number of SCS per month and the comparison of GPs with varying lengths of experience do not reach the desired sample size per group. The low sample sizes in subgroup analyses limits the statistical representativeness and generalizability. Concerning the validity of the survey, the content validity is given:

GPs were asked about their screening behavior that represents the quality of SCS. The survey might be biased by the selection of the sample and the memories of the participants (recall bias). Furthermore, social desirability bias might play a role here. Accordingly, socially desirable answers are given, for example that all body regions are always examined during screening, although this is not the case in reality. Since Görig et al. [22] yielded similar results in some categories, this study can contribute a part to the criterion validity. Still, criterion validity is difficult to achieve because there is no other test recognized as valid that could be consulted to the same GPs analyzed here to check the quality of SCS in GPs´ offices.

## Conclusions

The GPs surveyed provided information on how they perform SCS, how they educate their patients, and provided further information on uncertainties, knowledge gaps, structural and organizational requirements of SCS, opinions on SCS training and evaluation. Significant differences were found according to the number of monthly screenings and age over 53 years. The survey showed that the entire skin is not always inspected. Comments on the open questions indicated that in daily routine, the requirement for inspections of certain areas of the body, such as genitalia, is not met. In this case, cooperation with other specialists would be useful. In addition, patient education should include instructions for self-inspection of the skin. Inconsistencies in organization and payment are a potential barrier to SCS implementation for GPs. In several cases, the respondents mentioned uncertainties regarding unclear findings and the use of the dermatoscope. These uncertainties should definitely be addressed in the training programmes. Concerning the training programme it should be discussed whether refresher courses regarding SCS especially for GPs would be useful and whether courses specifically targeted for subgroups, for example young professionals, should be offered in order to improve the SCS in the long term [26].

## Supporting information

**S1 Appendix. STROBE checklist.** Report according to the requirements of the STROBE guidelines (https://www.strobe-statement.org/).
(DOCX)

**S2 Appendix. Questionnaire.** Survey on skin cancer screening from the perspective of general practitioners.
(DOCX)

**S3 Appendix. Dataset.**
(XLSX)

**S4 Appendix. Categories.** Categories of qualitative analysis generated from the responses of general practitioners.
(DOCX)

## Author Contributions

**Conceptualization:** Lydia Reinhardt, Cristin Strasser, Theresa Steeb, Markus V. Heppt.

**Data curation:** Anne Petzold.

**Investigation:** Lydia Reinhardt, Cristin Strasser, Theresa Steeb.

**Methodology:** Lydia Reinhardt, Cristin Strasser, Theresa Steeb, Anne Petzold.

**Project administration:** Lydia Reinhardt, Cristin Strasser, Theresa Steeb.

**Supervision:** Carola Berking, Friedegund Meier.

**Writing – original draft:** Lydia Reinhardt, Cristin Strasser.

**Writing – review & editing:** Lydia Reinhardt, Cristin Strasser, Theresa Steeb, Anne Petzold, Markus V. Heppt, Anja Wessely, Carola Berking, Friedegund Meier.

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
