## [Decision Letter · Decision Letter 0]

12 Sep 2023

PONE-D-23-14507General practitioners' perspectives on statutory skin cancer screening – a questionnaire-based cross-sectional survey in GermanyPLOS ONE

Dear Dr. Reinhardt,

Thank you for submitting your manuscript to PLOS ONE. After careful consideration, we feel that it has merit but does not fully meet PLOS ONE’s publication criteria as it currently stands. Therefore, we invite you to submit a revised version of the manuscript that addresses the points raised during the review process.

Please find attached two reviews that may guide your revision. Please pay high attention to the individual concerns, particularly with the extent of validity of your findings.

We look forward to receiving your revised manuscript.

Kind regards,

Felix G. Rebitschek

Academic Editor

PLOS ONE

Journal Requirements:

Reviewers' comments:

Reviewer's Responses to Questions

**Comments to the Author**

1. Is the manuscript technically sound, and do the data support the conclusions?

Reviewer #1: Yes

Reviewer #2: Yes

2. Has the statistical analysis been performed appropriately and rigorously? 

Reviewer #1: Yes

Reviewer #2: Yes

3. Have the authors made all data underlying the findings in their manuscript fully available?

Reviewer #1: Yes

Reviewer #2: Yes

4. Is the manuscript presented in an intelligible fashion and written in standard English?

Reviewer #1: Yes

Reviewer #2: Yes

5. Review Comments to the Author

Reviewer #1: Dear authors,

thank you very much for this interesting work on skin cancer screening. Considering the GPs perspective on the German statutory skin cancer screening is an important topic in order to also be able to assess the quality of the screening and derive future recommendations. Therefore, I rate the presented study positively, but think that it needs some major revision before publication. Overall the paper remains facile, specially in discussion and conclusion and the following comments should be considered:

- Information on the validity of the study should be provided. Even though representativeness is not sought, comments on needed power and sample size would be good. For assessing the informative value of the study, providing numbers of the total population in terms of age, gender and, if possible, years of experience for both regions to relate them to the study participants would be necessary. In addition, it would be important to know whether the characteristics mentioned differ between the two study regions in Bavaria and Saxony. The limited statistical representativeness should also be discussed.

- In the subgroup comparisons, the mean ranks should be given.

- The qualitative results presented appear to be more of a summary of the comments mentioned and not really a dedicated content analysis. A revision should take place here.

- The discussion should be revised and be more comprehensively. It would be very useful to compare the results of skin cancer screening from the patient's point of view with the subjective perception of the GPs considering current studies on this (see e.g. Görig et al 2021). Overall, the significant results of the subgroup differences remain unconsidered.

- Since the 8-hour training is mentioned several times, some background information should be provided. For example, that the GPs and dermatologists complete the same training, that the content of the training is determined by the KFR and what content is taught there.

In addition, the conclusion demands that uncertainties and the use of a dermatoscope should be addressed in the training, which is both already part of the training, as far as I know. In addition, a refresher course and an evaluation are recommended. Such an evaluation has always existed and there was also a refresher, but it was discontinued due to low demand. Therefore, a mandatory refresher specified by the KFE may be more useful to be discussed.

Reviewer #2: This is a well-written manuscript on the perspective of skin cancer prevention programs, which are essentially carried and implemented by primary care physicians. It can be assumed that due to the high quantitative demand the care of the people would not be guaranteed by the dermatologists alone. Already in 2014, a Schäfer et al. showed that tumor thickness is significantly higher in rural areas, in areas where specialist care is critical and will be increasingly so (1). Against this background, the present work is of political interest even if the geographical area as well as the participation is limited, but should emphasize this importance even more.

Introduction

- Are there any data about the number of performed screenings in Saxony and Bavaria by either general practitioners or dermatologists? This would be an interesting information.

Methods

Methods and statistics are described properly.

Discussion and conclusion

- The low participation of physicians has already been mentioned, and presumably physicians critical of screening in particular did not participate. This should be taken into account.

- Where possible, data on the frequency of screening in the two states should be related to the physicians surveyed and their proportion of screenings.

In summary, the paper addresses the perspective of general practitioners towards Skin Cancer Screening. Due to the geographic selection of two metropolitan regions, this is not a fully representative sample, but it nevertheless provides interesting information about the perspective of general practitioners.

(1) Schäfer I, Reusch M, Siebert S, Spehr C, Augustin M: Versorgungsmerkmale des Basalzellkarzinoms in Deutschland: Die Rolle von Versichertenstatus und sozio-demographischen Faktoren. J Dtsch Dermatol Ges. 2014

6. PLOS authors have the option to publish the peer review history of their article (what does this mean?). If published, this will include your full peer review and any attached files.

Reviewer #1: No

Reviewer #2: No

---

## [Author Response · Author response to Decision Letter 0]

26 Oct 2023

Thank you very much for the valuable comments and the opportunity to improve and revise our manuscript. Please find our answers in the rebuttal letter. We have tried to address all the reviewers’ concerns in a proper way and believe that our paper has improved considerably. We would be happy to make further corrections if necessary and look forward to hearing from you soon.

---

## [Decision Letter · Decision Letter 1]

5 Feb 2024

PONE-D-23-14507R1General practitioners' perspectives on statutory skin cancer screening – a questionnaire-based cross-sectional survey in GermanyPLOS ONE

Dear Dr. Reinhardt,

Thank you for submitting your manuscript to PLOS ONE. After careful consideration, we feel that it has merit but does not fully meet PLOS ONE’s publication criteria as it currently stands. Therefore, we invite you to submit a revised version of the manuscript that addresses the points raised during the review process.

We look forward to receiving your revised manuscript.

Kind regards,

Felix G. Rebitschek

Academic Editor

PLOS ONE

Additional Editor Comments:

Thank you very much for your submission with important results about how GPs inform patients and how they conduct dermatology standard examinations in the real world, and for addressing the reviews!

A couple of issues yet need more elaboration.

The following refers to the revision with track changes:

Major:

- It is not clear why you study what you study. Please show how you derive

o why you study, how the GPs perform/conduct the screening sessions, and, similarly,

o why you study their patient information strategies (what is here the desirable norm for comparison, are there deviations from informed screening decision making)?

o why you study their attitudes (evaluation)?

- The method leaves lacks that need to be filled for some reproducibility

o It is very good that the questionnaire is attached. The section “survey”, typically “Measures”, should nevertheless explain better what they different parts are about. This should be comprehensible without the appendix. Was this instrument validated, or used before? Was it pre-tested? If not, how it was developed? Also, how were qualitative data measured?

o How did you recruit with the help of the contact list (mailings)? How it was conducted (paper-and-pencil)? Was the procedure fixed, or could they scroll over the full survey back and forth?

o Qualitative analysis plan is underspecified; for instance I cannot infer the measures for assuring quality; please check PLOSONE here: Qualitative research studies should be reported in accordance to the Consolidated criteria for reporting qualitative research (COREQ) checklist or Standards for reporting qualitative research (SRQR) checklist.

o Because it is a survey study that aims to represent experts in Germany, we should get an indication of

How successful was recruitment, given population; recruitment and eligibility flowchart would be very helpful.

How representative was your sample (table 1), compared with the population of GPs in those states, or at least, in Germany, e.g. in terms of gender (an additional column with those data would be useful)?

- One can stress that the key results lie in the compliance with how to inform patients prior to examination (e.g., guidelines on evidence-based health communication) and how to conduct skin cancer examinations for purpose of screening (e.g., dermatology guidelines). Accordingly,

o You could also change Figure 1 to the analysis of proportions (I always…) This would be consistent to Figure 2 and supportive to the function of a visual (“patterns”) – complex ranking data as here could go with a table.

o Calculate (particularly with regard to further subgroup analysis) the proportions of those, who do not fully inform, and those who do not fully conduct. This stresses the need for intervention as you discuss it already.

- Subgroups: Because there was no preregistration and particularly no statistical hypotheses, please adjust alpha for multiple testing (depending on the number of your comparisons there). Furthermore, I would suggest to conduct those subgroup analyses preferably with the proportion of those who “always” did something (more robust!)

- Check reporting of qualitative results against Consolidated criteria for reporting qualitative research (COREQ) checklist or Standards for reporting qualitative research (SRQR) checklist, e.g. how many said something “was described as not relevant”…?

Discussion

- This should come back to the arguments that underlie your study motivation (see first point above).

Minors

- Suggestion to generally keep the order of your results section across the manuscript, first writing about “Informing patients” and then “Performing screening” (as it would be a the GP’s office)

- Consistent capitalisation across tables and figures and their elements please

- Consistent use of SCS across Figure captions and manuscript

- There are very important statements which do not have a reference, but which are result of the study, nor general knowledge

o L.53,54

o L.61/62

o L. 263/264

o L. 269 – could you check real-world materials for this aspect?

Abstract

- available = available free of charge?

- Objective: How = how comprehensive

- What other issues… This needs to be more concrete. What did you want to find out? How they inform patients?

Introduction:

- Do ref 2,3 control for screening programme introductions?

- Please add a sentence (based on the best available evidence) about benefits and harms of skin cancer screening for the general population from 35 years

- L.70-73, please provide the baseline numbers (of GPs and dermatologists)

Methods

- Please check order of sections against PLOSONE guidelines; also study population could go to the Methods

- Funding statement comes separately

- Delta is between absolute means, if so, expected effect size? To which outcome was this expectation referred?

Results

- L.160 , n=1?

- For your quantitative sample-based results from which you would like to generalise in the discussion, you need to provide uncertainty information (e.g. confidence intervals)

Discussion

- L. 319 “in which the uncertainties…” could be skipped here,because this is only a sub-aspect

- The construct validity does not depend on the sample; explain why recall bias affect construct validity?

Reviewers' comments:

Reviewer's Responses to Questions

**Comments to the Author**

1. If the authors have adequately addressed your comments raised in a previous round of review and you feel that this manuscript is now acceptable for publication, you may indicate that here to bypass the “Comments to the Author” section, enter your conflict of interest statement in the “Confidential to Editor” section, and submit your "Accept" recommendation.

Reviewer #1: All comments have been addressed

Reviewer #3: (No Response)

2. Is the manuscript technically sound, and do the data support the conclusions?

Reviewer #1: Yes

Reviewer #3: Yes

3. Has the statistical analysis been performed appropriately and rigorously? 

Reviewer #1: Yes

Reviewer #3: Yes

4. Have the authors made all data underlying the findings in their manuscript fully available?

Reviewer #1: Yes

Reviewer #3: No

5. Is the manuscript presented in an intelligible fashion and written in standard English?

Reviewer #1: Yes

Reviewer #3: Yes

6. Review Comments to the Author

Reviewer #1: Thank you very much for your response and the consideration of my comments. They have all been addressed adequatly.

Reviewer #3: I am coming to this manuscript as a new reviewer following revision by the authors. The authors appear to have addressed the comments from the previous reviews in a thorough and satisfactory manner. I have one minor comment, which is that it would be helpful to understand whether the training explicitly addresses which areas of the body should be screened for skin cancer. The authors highlight their findings about which areas of the body are less frequently screened by survey respondents, but they also mention that skin cancer is less frequent in those areas. It would be useful to understand the degree to which this is part of the training.

My other comment is on data availability. The authors mention have made the survey instrument and categorization scheme available, but this is insufficient to reproduce the results. If it is possible to make an anonymized file of survey responses available, the authors should do so, or explain the restrictions that don't allow them to make this data available.

7. PLOS authors have the option to publish the peer review history of their article (what does this mean?). If published, this will include your full peer review and any attached files.

Reviewer #1: No

Reviewer #3: No

---

## [Author Response · Author response to Decision Letter 1]

25 Mar 2024

We would like to express our gratitude to you for providing us with valuable feedback. Please find our answers in the rebuttal letter. We have made every attempt to fully address your comments. We hope that these revisions meet the requirements for publication in the PLOS ONE Journal. Please let us know if any further changes are necessary.

---

## [Editor Report · Decision Letter 2]

18 Apr 2024

PONE-D-23-14507R2General practitioners' perspectives on statutory skin cancer screening – a questionnaire-based cross-sectional survey in GermanyPLOS ONE

Dear Dr. Reinhardt,

Thank you for submitting your manuscript to PLOS ONE. After careful consideration, we feel that it has merit but does not fully meet PLOS ONE’s publication criteria as it currently stands. Therefore, we invite you to submit a revised version of the manuscript that addresses the points raised during the review process.

We look forward to receiving your revised manuscript.

Kind regards,

Felix G. Rebitschek

Academic Editor

PLOS ONE

Journal Requirements:

Additional Editor Comments:

Thank you for your substantial and detailed revision!

One point remains concerning the Figures 2 and 3.

As I pointed out before about text. If the authors want to report conclusions about the population of GPs practising SCS, sample-based data should not be reported without uncertainty estimates in the figures. Without those clarifications figures could be easily misinterpreted.

If this is addressed, the manuscript can be published.

---

## [Author Response · Author response to Decision Letter 2]

14 Jul 2024

We would like to thank you for your constructive and detailed comments on our manuscript. The recommendations and advice have helped us to improve the quality of the manuscript significantly. As suggested, we have revised Figures 2 and 3. We hope that our resubmission is now suitable for inclusion in PLOS ONE and we look forward to hearing from you.

---

## [Editor Report · Decision Letter 3]

25 Jul 2024

General practitioners' perspectives on statutory skin cancer screening – a questionnaire-based cross-sectional survey in Germany

PONE-D-23-14507R3

Dear Dr. Reinhardt,

We’re pleased to inform you that your manuscript has been judged scientifically suitable for publication and will be formally accepted for publication once it meets all outstanding technical requirements.

Kind regards,

Felix G. Rebitschek

Academic Editor

PLOS ONE
---

## [Editor Report · Acceptance letter]

30 Jul 2024

PONE-D-23-14507R3 

PLOS ONE

Dear Dr. Reinhardt, 

I'm pleased to inform you that your manuscript has been deemed suitable for publication in PLOS ONE. Congratulations! Your manuscript is now being handed over to our production team.

Kind regards, 

on behalf of

Dr. Felix G. Rebitschek 

Academic Editor

PLOS ONE